# Redescription and First Nucleotide Sequences of *Opecoeloides pedicathedrae* (Digenea: Opecoelidae), a Parasite of *Cynoscion leiarchus* (Cuvier, 1830) (Eupercaria: Sciaenidae) from Brazil

**Melissa Querido Cárdenas** [1], **Simone Chinicz Cohen** [1,*], **Amanda Gleyce Lima de Oliveira** [2], **Marcia Cristina Nascimento Justo** [1] and **Cláudia Portes Santos** [2]

1. Laboratório de Helmintos Parasitos de Peixes, Instituto Oswaldo Cruz, Fiocruz, Rio de Janeiro 21040-360, RJ, Brazil; melissaq@ioc.fiocruz.br (M.Q.C.); marciajusto@ioc.fiocruz.br (M.C.N.J.)
2. Laboratório de Avaliação e Promoção da Saúde Ambiental, Instituto Oswaldo Cruz, Fiocruz, Rio de Janeiro 21040-360, RJ, Brazil; amandagleycelo@gmail.com (A.G.L.d.O.); portesclaudia@gmail.com (C.P.S.)
* Correspondence: cohen.simone@gmail.com

**Abstract:** *Opecoeloides* Odhner, 1928, is represented by 19 valid species found in marine fish, of which five have been reported in Brazil. Specimens of *Opecoeloides pedicathedrae* Travassos, Freitas & Bührnheim, 1966, were collected from the intestine of smooth weakfish *Cynoscion leiarchus*, a new host record, from off the coast of Rio de Janeiro, Brazil. They were examined using light and confocal laser microscopy. New partial sequences of 18S and 28S rDNA genes of *O. pedicathedrae* were obtained. Bayesian inference analysis on the partial 28S rDNA dataset resulted in a phylogram in which *O. pedicathedrae* formed a well-supported clade with *Opecoeloides fimbriatus* and *Opecoeloides furcatus*. The K2p distance between *O. pedicathedrae* and *O. fimbriatus* was 0.34%, with 3 divergent nucleotides; and between *O. pedicathedrae* and *O. furcatus* was 4.18%, with 38 divergent nucleotides. A Bayesian-inference phylogenetic tree based on the 18S rDNA recovered two main clades with five subfamilies. A clade of Opecoelinae showed that *O. pedicathedrae* was closer to *Pseudopecoeloides tenuis*; the K2p distance between these species was 2.14%, with 28 divergent nucleotides. The new nucleotide sequences presented inclusion of a phylogenetic analysis that can help to clarify the understanding of this complex taxon.

**Keywords:** Opecoelinae; Trematoda; digenean; *Opecoeloides*; 18S and 28S rDNA; phylogeny





## 1. Introduction

The Opecoelinae Ozaki, 1925, represents one of the largest subfamilies within the Opecoelidae; it is characterized by bearing a cirrus sac that is reduced or absent, and there is no canalicular seminal receptacle [1]. This subfamily comprises 21 valid genera [2], including *Opecoeloides* Odhner, 1928, which is represented by 19 valid species found in marine fish, of which 5 have been reported in Brazil: *Opecoeloides catarinensis* Amato, 1983; *Opecoeloides melanopteri* Amato, 1983; *Opecoeloides pedicathedrae* Travassos, Freitas & Bührnheim, 1966; *Opecoeloides polynemi* Von Wicklen, 1946; and *Opecoeloides stenosomae* Amato, 1983 [2–4].

During a survey on parasites infecting *Cynoscion leiarchus* (Cuvier), commonly known as the smooth weakfish, which was caught off the coast of Rio de Janeiro, Brazil, specimens of *O. pedicathedrae* were collected. This pelagic fish species occurs in the western Atlantic Ocean, from Massachusetts, USA, down to Rio de Janeiro, Brazil. It feeds on other fish, mollusks, and crustaceans [5], and is an important regional fish resource [6].

*Opecoeloides pedicathedrae* was first described by Travassos et al. [3] from the intestine of the sand drum, *Umbrina coroides* Cuvier, off the state of Espírito Santo, Brazil. Subsequently, it was reported from the intestine of the Southern kingcroaker, *Menticirrhus americanus*

(Linnaeus), the flying gurnard *Dactylopterus volitans* (Linnaeus), and the barbel drum *Ctenosciaena gracilicirrhus* (Metzelaar), off the Rio de Janeiro coast, Brazil [7–10]. In the present study, new light microscopy and confocal laser microscopy analysis on *O. pedicathedrae* collected from *C. leiarchus* and new partial sequences of 18S and 28S rDNA genes are provided. The new nucleotide sequences presented, together with phylogenetic analyses, will help to clarify the understanding of this complex taxon.

## 2. Materials and Methods

### 2.1. Specimen Collection and Morphological Analysis

During October 2021, one specimen of *C. leiarchus* was bought from the fish market of Copacabana, off the coastal zone of Rio de Janeiro, Brazil (22°58′15″ S, 43°10′54″ W). The collection for this study was authorized by the Biodiversity Authorization and Information System (SISBIO, number 44652-5). The fish was kept in ice and immediately brought to the laboratory for analysis and identified according to Froese and Pauly [11]. Digeneans were collected alive from the intestine, washed in saline solution, and fixed in alcohol 70% under light cover glass pressure and without pressure. The specimens were stained with Langeron's alcoholic acid carmine, dehydrated by means of an ethyl alcohol series, cleared using Clove Oil, and mounted in Canada balsam as permanent slides. Photomicrographs were taken using a Zeiss® Axioskop microscope micrographic system with a differential interference contrast (DIC) apparatus and a confocal laser scanning with a ZEISS LSM 510 microscope. The measurements of compressed specimens followed by uncompressed specimens in brackets are given in micrometers unless otherwise stated, and the range is presented followed by the mean in parentheses. The measurements of compressed specimens are also provided in Table 2 for comparative purposes with previous descriptions. Specimens deposited in the Helminthological Collection of the Instituto Oswaldo Cruz (CHIOC) under numbers 29,977 and 29,976 by Travassos et al. [3] were investigated.

### 2.2. DNA Extraction, PCR Amplification, and DNA Sequencing

DNA extraction was performed using a QIAamp DNA Mini Kit (QIAGEN) according to the manufacturer's instructions, and a set of primers were used to amplify different regions of the DNA. The rDNA region 28S was amplified by PCR using the C1 (5′-ACCCGCTGAATTTAAGCAT-3′) and D2 (5′-TGGTCCGTGTTTCAAGAC-3′) primers using cycling parameters as stated by Chisholm et al. [12]. For partial 18S rDNA, the Het 18SF (5′-TCATATGCTTGTCTCAGA-3′) and Het 18SR (5′-ACGGAAACCTTGTTACGA-3′) primers [13,14] were used. PCRs for the 28S region were carried out using the following cycling parameters: initial denaturation step at 94 °C (2 min), followed by 40 cycles at 95 °C (30 s), 55 °C (30 s), and 72 °C (60 s), and a final extension step at 72 °C (5 min). For the partial 18S region, the reaction was carried out under the following conditions: 95 °C for 5 min (initial denaturation), followed by 40 cycles at 94 °C for 30 s, 53 °C for 30 s, and 72 °C for 45 s, and 72 °C for 7 min [12–14]. The PCR products were analyzed by electrophoresis in 1.5% agarose in Tris-borate EDTA gels, stained with SyberGreen (Invitrogen, Eugene, OR, USA), and photographed under UV transillumination. The amplified PCR products were purified using ExoSap-IT (USB® Products Affymetrix Inc., Cleveland, OH, USA). DNA cycle sequencing reactions were performed using the BigDye Terminator v.3.1 (Applied Biosystems, Foster City, CA, USA), and automated sequencing was performed using the Sequencing Platform at the Fundação Oswaldo Cruz (PDTIS/Fiocruz) in Brazil. Sequences of both strands were generated, edited, and aligned by using the MEGA software version 11 [15]. The sequences were compared to others available in the GenBank database using the Basic Local Alignment Search Tool (BLAST) program from the National Center for Biotechnology Information (NCBI) server (http://www.ncbi.nlm.nih.gov/BLAST (accessed 8 December 2022) [16]. Evolutionary divergence estimates between sequences were conducted in MEGA11 using the Kimura 2-parameter (K2p) model [17].

To examine phylogenetic relationships, nucleotide sequences were aligned using CLUSTAL W in the MEGA11. Bayesian inference phylogenetic trees were conducted using

Monte Carlo Markov Chain (MCMC) analysis available in the BEAST v2.6.3 software [18]. Likelihood parameters set for the BI analysis were based on the Akaike Information Criteria (AIC) test in jModelTest2 [19]. The selected model was the General Time-Reversible (GTR) for 28S, and the Hasegawa-Kishino-Yano (HKY) for the 18S, employing the birth–death model (BDM). Posterior probabilities (pp) were calculated via 10,000,000 generations, sampling every 1000th tree being saved. Tracer v1.7.2 [20] was used to validate the convergence and mixing to ensure all effective sample size (ESS) values greater than 200. The trees were presented as Maximum-Clade Credibility (MCC) trees using the TreeAnnotator v2.6.3 software after discarding the first 10% as burn-in and visualized using the FigTree v1.4.4 [20]. For tree rooting, the best sequences used as outgroups were *Enenterum aureum* Linton, 1910, and *Preptetos caballeroi* Pritchard, 1960, for both regions. The sequences from GenBank that were used for the phylogenetic analysis are listed in Table 1.

**Table 1.** Digeneans used in the phylogenetic analyses with their respective GenBank accession numbers.

| Opecoelidae | 28S rDNA | 18S rDNA | Reference |
|---|---|---|---|
| **Opecoelinae** | | | |
| *Opecoeloides furcatus* (Bremser in Rudolphi, 1819) | AF151937 | - | [21] |
| *Opecoeloides fimbriatus* (Linton, 1934) | KJ001211 | - | [22] |
| *Opecoeloides fimbriatus* (Linton, 1934) | MK648309 | | [23] |
| *Anomalotrema koiae* Gibson & Bray, 1984 | KU320595 | KU320582 | [24] |
| *Pseudopecoeloides tenuis* Yamaguti, 1940 | KU320605 | KU320592 | [24] |
| *Pseudopecoelus vulgaris* (Manter, 1934) | MH161436 | - | [25] |
| *Discoverytrema gibsoni* Zdzitowiecki, 1990 | MH161430 | - | [25] |
| *Discoverytrema markowskii* Gibson, 1976 | MH161431 | - | [25] |
| *Dimerosaccus oncorhynchi* (Eguchi, 1931) | FR870262 | - | [26] |
| **Helicometrinae** | | | |
| *Helicometra fasciata* (Rudolphi, 1819) | KU320597 | KU320584 | [24] |
| *Helicometra boseli* Nagaty, 1956 | KU320600 | KU320587 | [24] |
| **Scorpidotrematinae** | | | |
| *Holsworthotrema enboubalichthys* Martin, Huston, Cutmore & Cribb, 2018 | MK052937 | MK052940 | [27] |
| *Holsworthotrema chaoderma* Martin, Huston, Cutmore & Cribb, 2018 | MK052938 | MK052941 | [27] |
| *Scorpidotrema longistipes* Aken'Ova & Cribb, 2003 | MK052936 | - | [27] |
| **Podocotylinae** | | | |
| *Buticulotrema thermichthysi* Bray, Waeschenbach, Dyal, Littlewood & Morand, 2014 | KF733984 | - | [28] |
| *Podocotyle atomon* (Rudolphi, 1802) | MH161437 | - | [25] |
| *Bathypodocotyle margolisi* (=*Allopodocotyle margolisi*) (Gibson, 1995) | KU320596 | KU320583 | [24] |
| *Halosaurotrema halosauropsi* (=*Gaevskajatrema halosauropsi*) (Bray & Campbell, 1996) | AY222207 | AJ287514 | [29,30] |
| **Bathycreadiinae** | | | |
| *Bathycreadium elongata* (Maillard, 1970) | JN085948 | - | [31] |
| **Plagioporinae** | | | |
| *Neoplagioporus ayu* (Takahashi, 1928) | KX553947 | - | [32] |
| *Neoplagioporus zacconis* (Yamaguti, 1934) | KX553949 | - | [32] |
| *Neoplagioporus elongatus* (Goto & Ozaki, 1930) | KX553948 | - | [32] |
| *Urorchis goro* Ozaki, 1927 | KX553946 | - | [32] |
| *Urorchis acheilognathi* Yamaguti, 1934 | KX553945 | - | [32] |
| *Sphaerostoma bramae* (Müller, 1776) | MH161435 | - | [25] |
| *Plagioporus fonti* Fayton, Choudhury, McAllister & Robison, 2017 | KX905054 | - | [32] |
| *Plagioporus boleosomi* (Pearse, 1924) | KX553953 | - | [32] |

**Table 1.** *Cont.*

| Opecoelidae | 28S rDNA | 18S rDNA | Reference |
|---|---|---|---|
| *Plagioporus sinitsini* Mueller, 1934 | KX553944 | - | [32] |
| *Plagioporus hageli* Fayton & Andres, 2016 | KX553950 | - | [32] |
| *Plagioporus loboides* (=*Plagiocirrus loboides*) (Curran, Overstreet & Tkach, 2007) (*Incertae sedis*) | EF523477 | - | [33] |
| **Subfamily Incertae sedis** | | | |
| *Abyssopedunculus brevis* (=*Podocotyloides brevis*) (Andres & Overstreet, 2013) | KJ001212 | - | [22] |
| *Mesobathylebouria lanceolata* (=*Neolebouria lanceolata*) (Price, 1934) | KJ001210 | - | [22] |
| **Outgroup** | | | |
| *Enenterum aureum* Linton, 1910 | AY222232 | AY222124 | [29] |
| *Preptetos caballeroi* Pritchard, 1960 | AY222236 | AJ287563 | [30] |

## 3. Results

*3.1. Morphological Analysis*

*Opecoeloides pedicathedrae* (Figures 1–3)

Host: *Cynoscion leiarchus* (Cuvier, 1830); Eupercaria: Sciaenidae; smooth weakfish.

Locality: Coastal zone off Rio de Janeiro, Brazil (22°58′15″ S, 43°10′54″ W).

Site of infection: Intestine.

Deposited specimens: Vouchers (n = 11) CHIOC 40259 a-d

Representative DNA sequences: 28S rDNA (GenBank accession no. MZ820429); 18S (GenBank accession no. OL998834).

Intensity of infection: one fish parasitized by 45 specimens.

Description based on 14 mature specimens: Body elongate, measuring 3.56–4.98 (4.45; n = 3) [1.78–2.54 (2.18; n = 6)] mm in length and 400–740 (520; n = 3) [240–670 (404; n = 6)] in width. Tegument smooth. Oral sucker [OS] opening subterminal, measuring 200–225 (210; n = 5) [155–175 (170; n = 6)] in length and 150–220 (187; n = 5) [125–225 (166; n = 6)] in width. Prepharynx short. Pharynx [Ph] oval, muscular, measuring 112–125 (116; n = 3) [75–140 (97; n = 4)] by 92–145 (114; n = 3) [75–140 (99; n = 4)]. Pharynx length (2.59–3.15%) [3.50–6.27%] of body length. OS: Ph length ratio 1: 1.76–2.00 (n = 3) [1.25–2.06; n = 3] and OS: Ph width ratio 1: 1.52–1.66 (n = 3) [1.35–1.82; n = 3]. Esophagus long. Intestinal bifurcation just anterior to accessory sucker, dividing into two caeca, which run in parallel toward posterior extremity, ending via uroproct. Ventral sucker [VS] located at anterior third of body, pedunculated, measuring 275–375 (335; n = 4) [210–350 (251; n = 10] in length and 235–300 (274; n = 4) [125–300 (192; n = 10)] in width. Oral sucker roughly half to third size of ventral sucker. VS:OS length ratio 1: 1.37–1.83 (n = 4) [1.28–2.0; n = 5] and VS:OS width ratio 1:1.18–1.58 n = 4 [1.2–1.6; n = 5]. Ventral sucker exhibiting ten muscular digitiform expansions with terminal papillae, disposed in two lateral groups, supported by muscle bundles. Accessory sucker strongly muscular, anteriorly to acetabular peduncle, measuring 82–115 (97; n = 4) [87–120 (97; n = 7)] in length and 85–145 (107; n = 4) [80–177 (98; n = 7)] in width. Testes round to oval, slightly indented, tandem, post-ovarian. Anterior testis 275–405 (342; n = 6) [145–305 (212; n = 9)] long by 215–330 (274; n = 6) [150–300 (213; n = 9)] wide, and posterior testis 310–435 (375; n = 5) [125–350 (259; n = 9)] long by 225–305 (277; n = 5) [135–325 (226; n = 9)] wide. In some specimens, space between testes filled by vitelline follicles. Seminal vesicle sigmoid, 160–360 (264; n = 4) long, opening in ejaculatory duct connected to a strongly muscular cirrus. Cirrus sac absent. Genital pore situated anteriorly to accessory sucker. Ovary oval, pre-testicular, 125–200 (159; n = 5) [120–175 (146; n = 4)] long and 180–280 (217; n = 5) [175–260 (219; n = 4)] wide. Vitelline follicles large, extending from proximal extremity of seminal vesicle to posterior end of body. Uterus pretesticular, intercaecal, extending between ovary and accessory sucker; distal region not differentiated into metraterm, passes dorsally or laterally to accessory sucker, and opens

into genital atrium. Eggs oval, operculate, 47–60 (54; n = 25) [40–62 (53; n = 25)] long by 25–37 (31; n = 25) [27–50 (37; n = 25)] wide. Excretory vesicle extends to ovary.

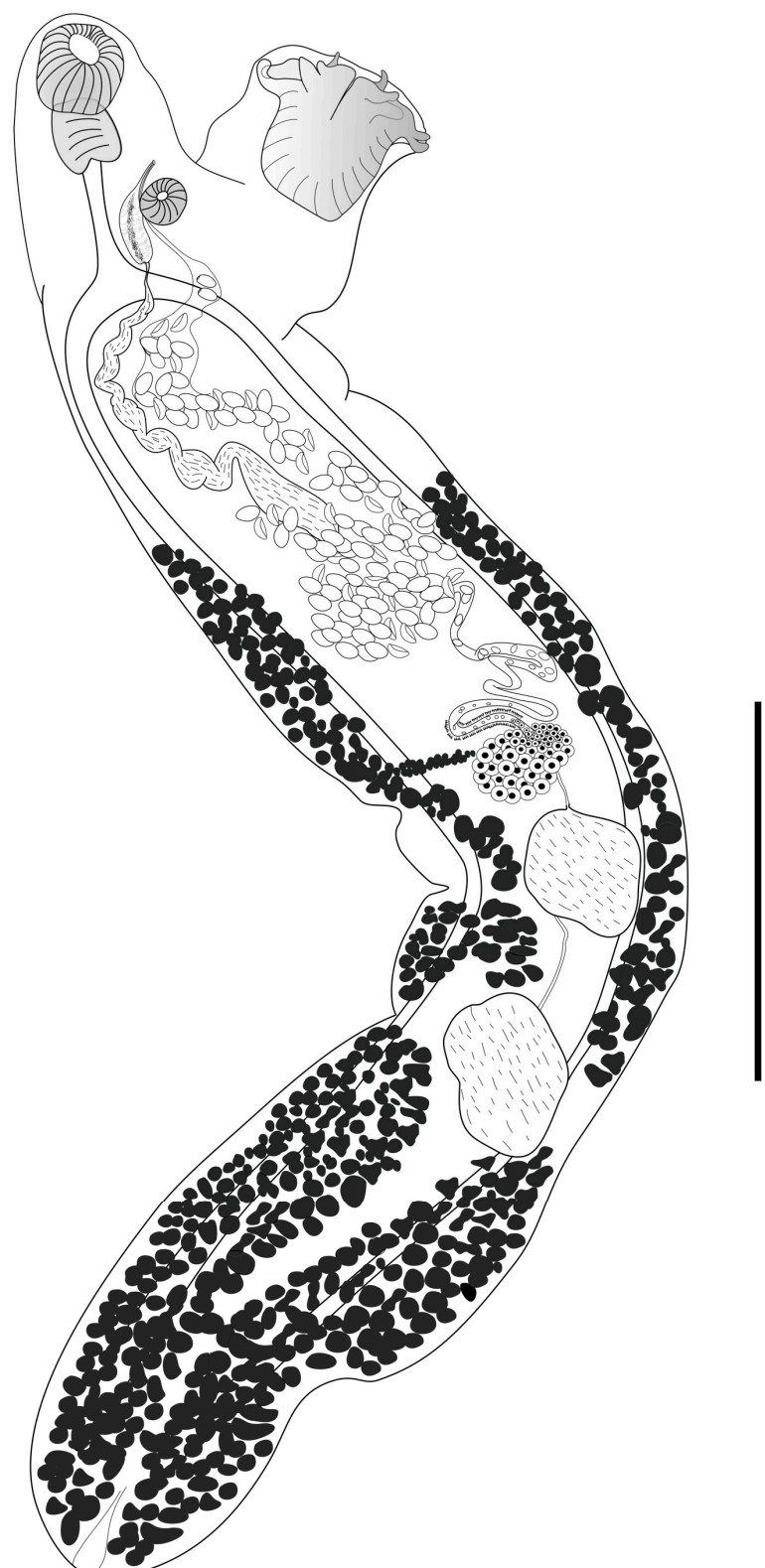

**Figure 1.** *Opecoeloides pedicathedrae* Travassos, Freitas & Bührnheim, 1966, from *Cynoscion leiarchus* from Brazil, line drawing. Bar 1 mm.

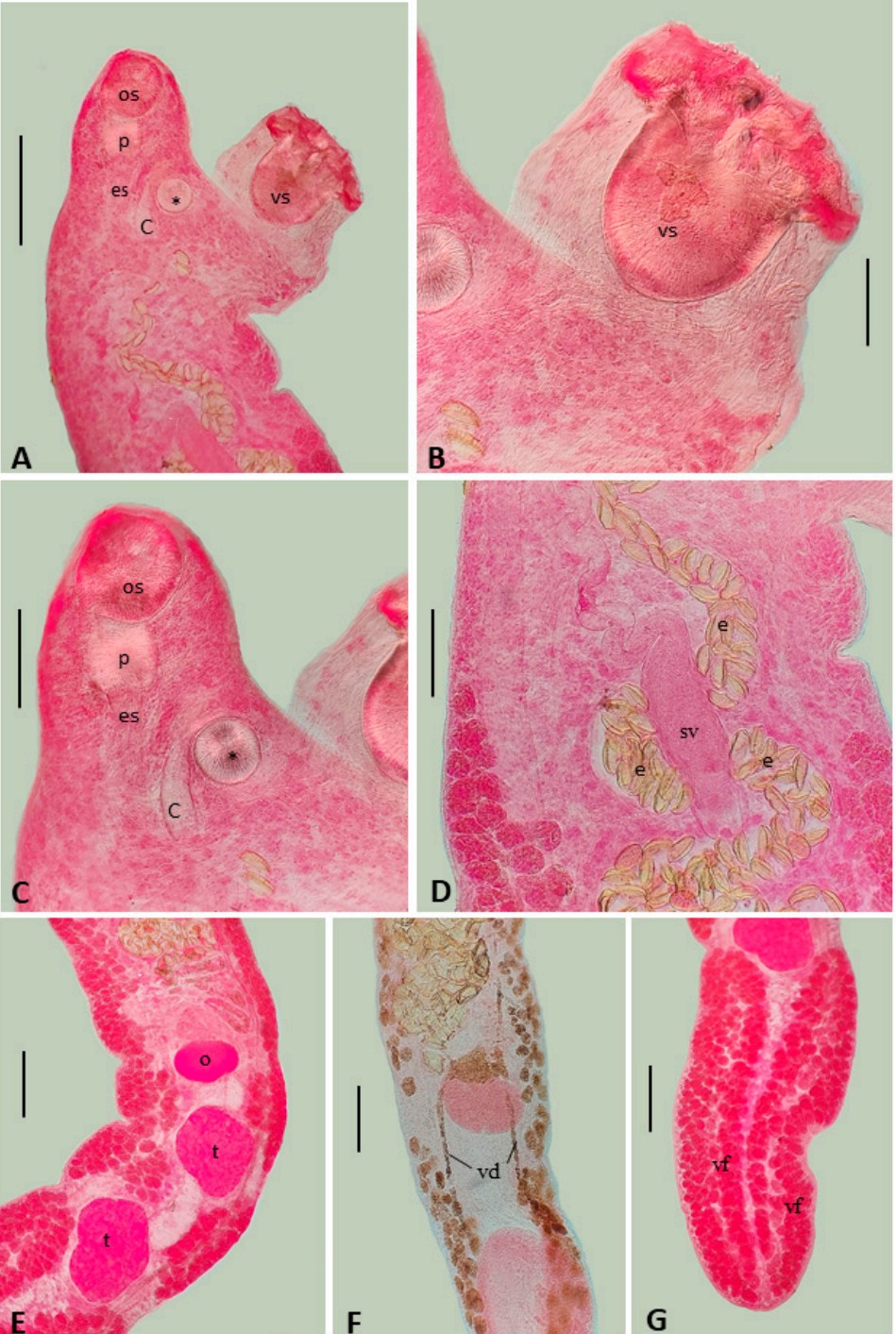

**Figure 2.** *Opecoeloides pedicathedrae* light microscopy micrographs. (**A**) Anterior region showing oral sucker (os), pharynx (p), esophagus (es), cirrus (c), accessory sucker (asterisk), and pedunculated ventral sucker (vs). (**B**) Detail of pedunculated ventral sucker (vs). (**C**) Detail of anterior region showing oral sucker (os), pharynx (p), esophagus (es), cirrus (c), and accessory sucker (asterisk). (**D**) Sigmoid seminal vesicle (sv) and eggs (e). (**E**) Midbody showing ovary oval (o) and testes slightly lobed in tandem (t). (**F**) Ovary (o), vitelline duct (vd), and testes (t). (**G**) Posterior region with vitelline follicles (vf) reaching to the end of the body. Bars: (**A**) = 400 μm; (**B**,**C**) = 200 μm; (**D**) = 150 μm; (**E**–**G**) = 350 μm.

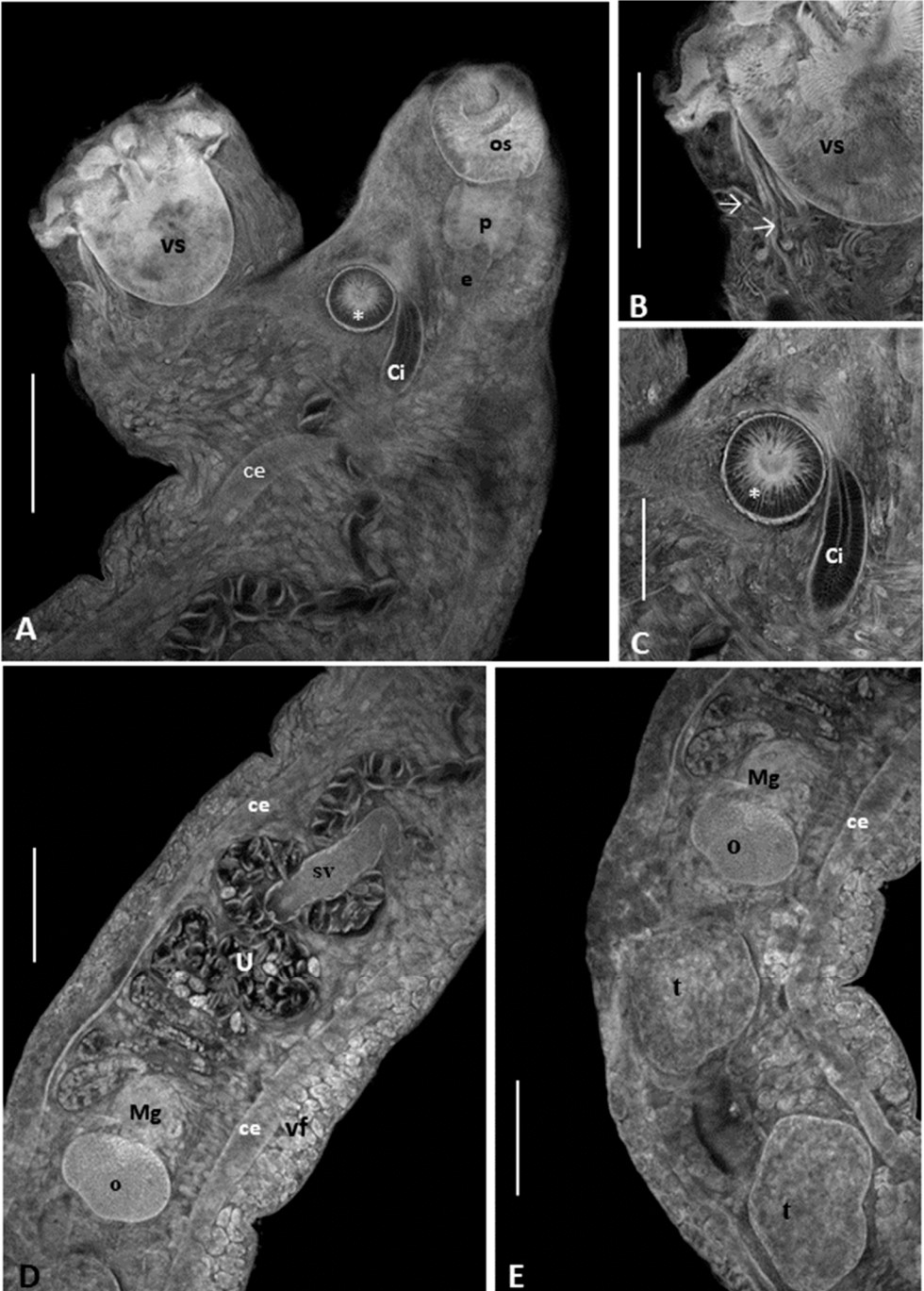

**Figure 3.** *Opecoeloides pedicathedrae* confocal laser scanning micrographs. (**A**) Anterior region showing oral sucker (os), pharynx (p), cirrus (ci), accessory sucker (asterisk), and pedunculated ventral sucker (vs). (**B**) Detail muscle bundles (arrow) supporting the ventral sucker (vs). (**C**) Detail of accessory sucker (asterisk) and cirrus (ci). (**D**) Midbody showing ovary oval (o), Mehlis' gland (Mg), uterus (U) with eggs, and seminal vesicle (sv). The caeca (ce) and vitelline follicles (vf) can also be observed. (**E**) Midbody showing ovary oval (o), Mehlis' gland (Mg), and testes slightly lobed in tandem (t). (ce) caeca. Bars: (**A,B,D,E**) = 200 μm; (**C**) = 100 μm.

Remarks

The comparative measurements of *O. pedicathedrae* from different host species are given in Table 2. The measurements of our specimens are in agreement with the original description of *O. pedicathedrae*. The only differences were the size of the eggs, which

were slightly larger (60–63 × 37–40 in the present study vs. 47–60 × 25–37 in the original description), and the larger widths of the ovary (290–330 in the present study vs. 180–280 in the original description) and testes (400–450 vs. 215–330 in the anterior testis and 400–450 vs. 225–305 in the posterior testis). The specimens of *O. pedicathedrae* studied by Fábio [7] and Fernandes et al. [8] from *M. americanus* also presented a smaller width of eggs and a larger range in the size of testis and ovary, thus demonstrating that there can be intraspecific variation in size according to the host.

**Table 2.** Comparative measurements of the present *Opecoeloides pedicathedrae* Travassos, Freitas & Buhrnheim, 1966 with other species described previously. Measurements are in micrometers except when indicated.

|  | Travassos et al. [3] | Fabio [7] | Fernandes et al. [8] | Present Study ** |
|---|---|---|---|---|
| Body | 2.37–6.56 × 0.82–0.83 mm | 4.25–4.9 × 0.45–0.78 mm | 2.35–6.83 × 0.26–0.71 mm | 3.56–4.98 × 400–740 mm |
| Oral sucker | 210–270 × 200–280 | 180–220 × 180–200 | 180–270 × 120–250 | 200–225 × 150–220 |
| Ventral sucker | 350–480 × 270–430 | 250–390 × 220–270 | 340–470 × 260–590 * | 275–375 × 235–300 |
| Ratio OS: VS | 1: 1.51–1: 1.65 | 1: 1.26–1: 1.57 |  | 1:1.37–1: 1.83 |
| Accessory sucker | 80–130 × 110–130 | 80–90 × 90 –100 | 0.6–0.12 × 0.07 × 0.13 mm | 82–115 × 85–145 |
| Pharynx | 110–160 × 100–130 | 110–120 × 110–140 | 0.08–0.15 × 0.08–0.12 mm | 112–125 × 92–145 |
| Seminal vesicle | 400 × 130 | 300–380 × 70–100 | 0.11–0.43 × 0.13 × 0.15 mm | 160–360 |
| Ejaculatory duct | 1120 × 400 |  |  |  |
| Cirrus | 270 × 70 |  |  |  |
| Anterior testis | 210–480 × 400–450 | 300–390 × 260–320 | 0.18–0.42 × 0.15–0.35 mm | 275–405 × 215–330 |
| Posterior testis | 270–480 × 400–450 | 330–340 × 280–320 | 0.20–0.47 × 0.16–0.38 mm | 310–435 × 225–305 |
| Ovary | 130–240 × 290–330 | 160–190 × 200–270 | 0.06–0.23 × 0.08–0.32 mm | 125–200 × 180–280 |
| Eggs | 60–63 × 37–40 | 52–66 × 31–46 | 0.059–0.068 × 0.026–0.034 mm | 47–60 × 25–37 |
| Host | *Umbrina coroides* | *Menticirrhus americanus* | *Menticirrhus americanus* | *Cynoscion leiarchus* |

* According to the authors, these measurements correspond to depth x longitudinal diameter; ** measurements based on compressed specimens; OS: oral sucker; VS: ventral sucker.

*3.2. Molecular Analysis*

In total, five sequences were obtained: two partial 18S rDNA (accession numbers OQ101905 and OQ101906, with 894 bp in each sequence) and three partial 28S rDNA sequences (OQ103465, OQ103466, and OQ103467; 921 bp, 906 bp, and 887 bp, respectively).

The first partial 18S rDNA sequence of *O. pedicathedrae* indicated 97.22% similarity to *Pseudopecoeloides tenuis* Yamaguti, 1940, and 97% to *Anomalotrema koiae* Gibson and Bray, 1984, both with 95% query cover. A Bayesian-inference phylogenetic tree based on the 18S rDNA recovered two main clades with five subfamilies. The Opecoelinae showed that *O. pedicathedrae* was closer to *P. tenuis* (92% probability) and separated from *A. koiae* (100% probability); the K2p distance between *O. pedicathedrae* and *P. tenuis* was 2.14%, with 28 divergent nucleotides in a 894 bp. The second main clade included the Podocotylinae Dollfus, 1959, together with Hamacreadiinae Martin, Downie & Cribb, 2020, followed by Scorpidotrematinae Sokolov, Shchenkov, Frolov & Gordeev, 2022, and separated from Helicometrinae Bray, Cribb, Littlewood & Waeschenbach, 2016 (Figure 4).

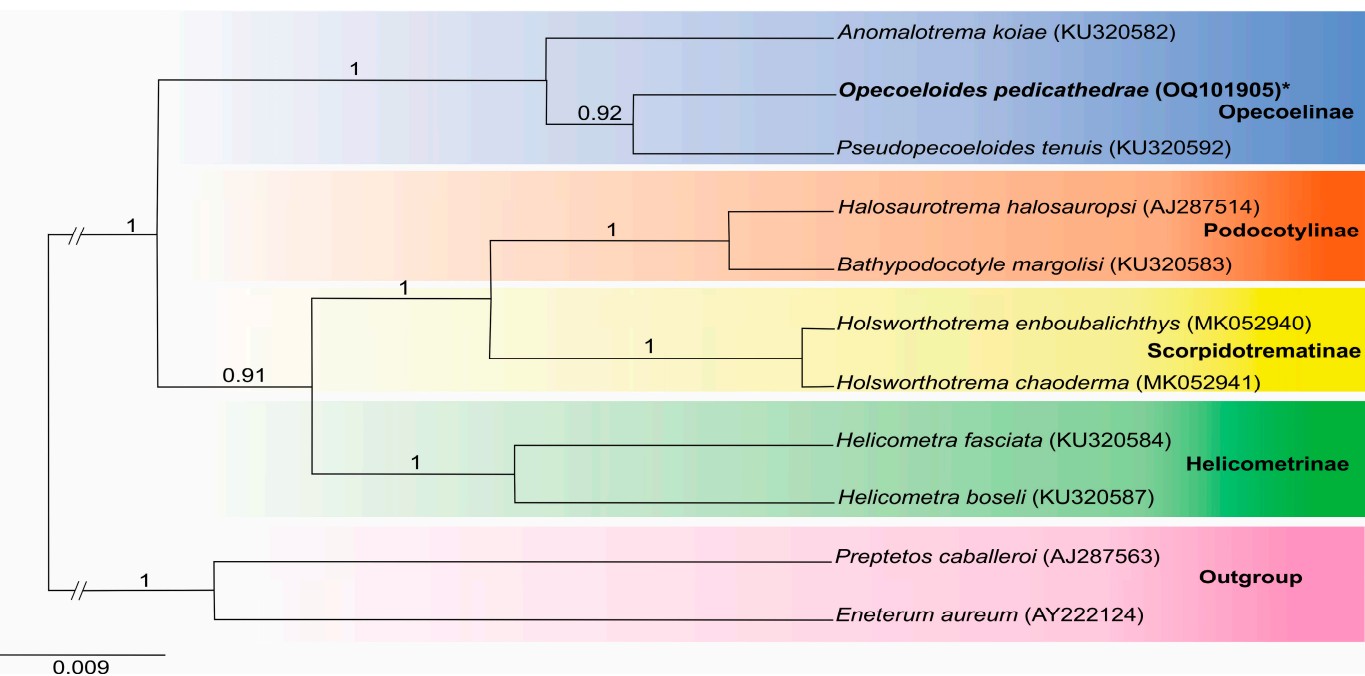

**Figure 4.** Bayesian phylogenetic topology with posterior probabilities indicating node support based on the 18S region to show their relationships with other species of Opecoelidae. The GenBank accession numbers are shown, and the scale bar indicates the nucleotide mutations per site. * New sequence data.

The partial 28S rDNA sequence of *O. pedicathedrae* showed similarities to *Opecoeloides fimbriatus* (Linton, 1934) with 99.66% similarity (97% query cover) and *Opecoeloides furcatus* (Bremser in Rudolphi, 1819) with 95.81% similarity (95% query cover). *Pseudopecoeloides tenuis* and *Pseudopecoelus vulgaris* (Manter, 1934) had 92.62% and 93.59% similarity, with 100% and 96% query cover, respectively. The K2p distance between *O. pedicathedrae* and *O. fimbriatus* (KJ001211) was 0.38%, with 3 divergent nucleotides, while the *O. fimbriatus* (MK648309) was 4.06%, with 32 divergent nucleotides; and the K2p distance between *O. pedicathedrae* and *O. furcatus* was 4.07%, with 34 divergent nucleotides in a 788 bp. The Bayesian-inference analysis on the partial 28S rDNA dataset resulted in a phylogram in which *O. pedicathedrae* formed a well-supported clade with *O. fimbriatus* and *O. furcatus*. With regard to the two sequences of *O. fimbriatus*, the specimens reported in the northern Gulf of Mexico were closer to *O. pedicathedrae* than the *O. fimbriatus* from Jalisco (Mexico). The clade containing species of *Opecoeloides*, *Pseudopecoeloides*, and *Pseudopecoelus* formed a clade of the Opecoelinae with *A. koiae*, *Discoverytrema gibsoni* Zdzitowiecki, 1990, *Discoverytrema markowskii* Gibson, 1976, and *Dimerosaccus oncorhynchi* (Eguchi, 1931). A separate main clade included species of the subfamilies Podocotylinae, Scorpidotrematinae, Plagioporinae, Bathycreadiinae, and Helicometrinae, while the species *Mesobathylebouria lanceolata* (Price, 1934) Martin, Huston, Cutmore & Cribb, 2018, and *Abyssopedunculus brevis* (Andres & Overstreet, 2013) Martin, Huston, Cutmore & Cribb, 2018, remained together in an *Incertae sedis* subfamily (Figure 5).

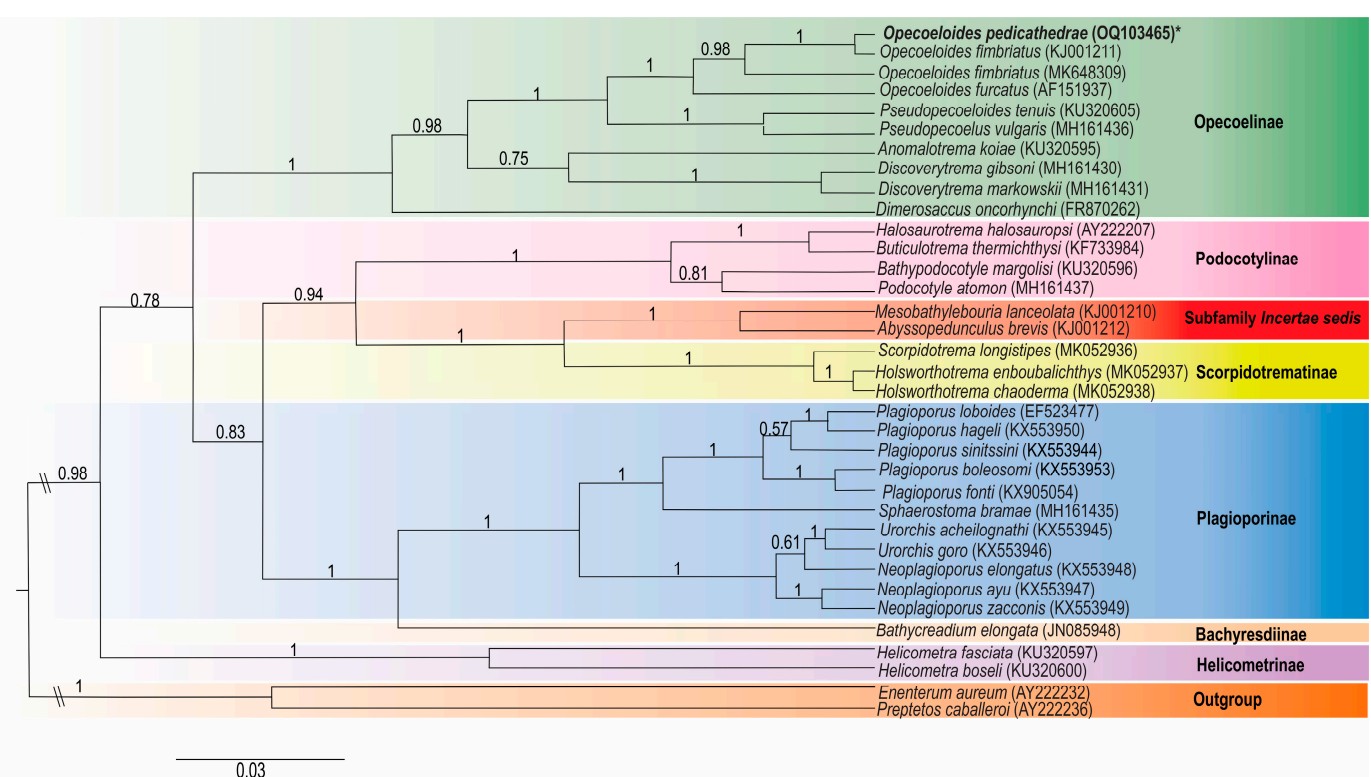

**Figure 5.** Bayesian phylogenetic topology with posterior probabilities indicating node support based on the 28S region to show their relationships with other species of Opecoelidae. The GenBank accession numbers are shown, and the scale bar indicates the nucleotide mutations per site. * New sequence data.

## 4. Discussion

Adult opecoelids live in the digestive tract of marine and freshwater fish and are mainly characterized by an aspinose tegument and a preovarian uterus. The genus *Opecoeloides* was proposed by Odhner [34] to accommodate *O. furcatus*. The species of this genus can be recognized by the combination of an "accessory sucker" immediately posterior to the genital pore, a caecum opening via an uroproct, the absence of a cirrus sac, and a ventral sucker that is usually pedunculate but may be protuberant [1].

The measurements of our specimens are in agreement with the original description of *O. pedicathedrae*. The only differences were the size of the eggs, which were slightly larger (60–63 × 37–40 in the present study vs. 47–60 × 25–37 in the original description), and the larger widths of the ovary (290–330 in the present study vs. 180–280 in the original description) and testis (400–450 vs. 215–330 in the anterior testis and 400–450 vs. 225–305 in the posterior testis). The specimens of *O. pedicathedrae* studied by Fábio [7] and Fernandes et al. [8] from *M. americanus* also presented a smaller width of eggs and a larger range in the size of testis and ovary, thus demonstrating that there can be intraspecific variation in size according to the host.

Cribb [1] recognized four subfamilies within the Opecoelidae: the Opecoelinae, the Plagioporinae Manter, 1937, the Stenakrinae Yamaguti, 1970, and the Opecoelininae Gibson & Bray, 1984. However, Bray et al. [24] demonstrated that the dominant opecoelid classification hypothesis at that time did not adequately reflect the phylogenetic relationships among taxa. Since then, studies based on a combination of adult morphology, host usage, and phylogenetic distinctions have accommodated new subfamily concepts, and Opecoelidae is currently divided into twelve subfamilies [24,35–37]. *Opecoeloides* was assigned to Opecoelinae, and two genera, *Abyssopedunculus* Martin, Huston, Cutmore & Cribb, 2018, and *Mesobathylebouria* Martin, Huston, Cutmore & Cribb, 2018, remain without suitable subfamilial designation [2,35–37]. Recently, Sokolov et al. [38] proposed a new subfamily,

the Scorpidotrematinae Sokolov, Shchenkov, Frolov & Gordeev, 2022, and removed the Stenakrinae from Opecoelidae, to recognize it as a separate family within Opecoeloidea, named Stenakridae Yamaguti, 1970.

The 18S and 28S rDNA nucleotide sequences of *O. pedicathedrae* have not previously been published. There are only 18S sequences from two *Opecoeloides* species in GenBank. However, the sequence of *O. furcatus* included few base pairs. The 18S rDNA tree showed that *O. pedicathedrae* and *P. tenuis* share a common clade that together with *A. koie* forms the Opecoelinae. A second clade includes three subfamilies: Podocotylinae closer to Scorpidotrematinae separated from Helicometrinae.

The 28S tree demonstrated that *O. pedicathedrae* is closer to *O. fimbriatus* reported in the northern Gulf of Mexico than the *O. fimbriatus* found in Jalisco, Pacific coast of Mexico. The partial 28S rDNA analysis demonstrated that the Opecoelinae, the Podocotylinae, the Scorpidotrematinae, the Plagioporinae, the Bathycreadiinae, and the Helicometrinae were organized in different clades with high statistical support, and the taxa of *A. brevis* and *M. lanceolata* also formed part of the same clade, thus corroborating previous phylogenetic hypotheses concerning Opecoelidae [24,25,35,36].

Bray et al. [24] hypothesized the monophyly of *Opecoeloides* spp. and *P. tenuis* by having an uroproct. However, their studies did not include *P. vulgaris*, which does not present an uroproct. Sokolov et al. [25] presented a phylogenetic tree of Opecoelinae with *O. fimbriatus* and *O. furcatus* in the same clade. The inclusion of the new sequences of *O. pedicathedrae* showed a phylogenetic convergence between *O. pedicathedrae* and *O. fimbriatus* compared to *O. furcatus*. *Opecoeloides pedicathedrae* was reported only off the coast of Brazil [3,7,8], while *Opecoeloides fimbriatus* was reported from the Western North Atlantic [22,39,40], both sides of the Panama Isthmus [41], and the Pacific coast of Mexico [23,42]. *Opecoeloides furcatus* was reported from Mediterranean waters [21,43,44], off the Canary Islands [45], off the coast of Belgium [46], and from Japan [47].

According to Bray et al. [24] the Opecoelinae is a well-supported group, both in terms of molecular and morphological evidence. Our findings demonstrate that *Opecoeloides* spp., which are characterized by having a pedunculated ventral sucker with terminal lobes and papillae, are separated from *Pseudopecoeloides* spp., which have a ventral sucker without any types of papillae, and *Pseudocoelus* spp., which presents a ventral sucker without prominent anterior and posterior lobes. Besides this, *Opecoeloides* spp. bear an accessory sucker, while other opecoelins lack such a structure.

Although molecular studies have begun to elucidate the complexity of Opecoelidae, a clear pattern has yet to be found [24]. This morphological redescription of *O. pedicathedrae* from a new host (*C. leiarchus*), with the first genetic study using 18S and 28S rDNA nucleotide sequences, could contribute to the understanding of the phylogeny of the family Opecoelidae.

**Author Contributions:** Conceptualization, M.Q.C. and C.P.S.; methodology, M.Q.C., S.C.C., A.G.L.d.O., M.C.N.J. and C.P.S.; investigation, M.Q.C., S.C.C., A.G.L.d.O., M.C.N.J. and C.P.S.; resources, M.Q.C. and C.P.S.; writing—original draft preparation, M.Q.C. and C.P.S.; writing—review and editing, S.C.C., A.G.L.d.O. and M.C.N.J.; supervision, M.Q.C., S.C.C. and C.P.S.; project administration, M.Q.C., S.C.C. and C.P.S.; funding acquisition, S.C.C. and C.P.S. All authors have read and agreed to the published version of the manuscript.

**Funding:** The present study was supported financially by the Oswaldo Cruz Foundation (PAEF no. IOC-023-FIO-18-2-4 and IOC-008-FIO-22-2-42).

**Institutional Review Board Statement:** Not applicable.

**Data Availability Statement:** Data are contained within the article.

**Conflicts of Interest:** The authors declare no conflict of interest.

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
