# Peer review of "Redescription and First Nucleotide Sequences of Opecoeloides pedicathedrae (Digenea: Opecoelidae), a Parasite of Cynoscion leiarchus (Cuvier, 1830) (Eupercaria: Sciaenidae) from Brazil"

_diversity, doi:10.3390/d16040197_

Round 1

Reviewer 1 Report

Comments and Suggestions for Authors

The manuscript diversity-2685029 provided the redescription of Opecoeloides pedicathedrae parasites from Cynoscion leiarchus (Cuvier, 1830) (Eupercaria: Sciaenidae) from Brazil. The paper is interesting. However, some suggestions are recommended that will make the work better articulated. Reminding the authors, before submitting the final version, to insert all the missing data relating to the deposited specimens.

Author Response

Answer to Reviewer 1

We would like to thank the reviewers for their valuable suggestions. The following changes were made according to their suggestions on the manuscript “Redescription and first nucleotide sequences of Opecoeloides pedicathedrae (Digenea: Opecoelidae), a parasite of Cynoscion leiarchus (Cuvier, 1830) (Eupercaria: Sciaenidae) from Brazil”. Most suggestions were accepted, and the exceptions are explained. A “Remarks” was added as suggested by Reviewer 1.

We have added the sequence of O. fimbriatus (MK648309) to the 28S phylogenetic tree as suggested by Reviewer 2, then some information has been added to the results and this sequence has been included in table 2. The suggested references were included.

REV 1

  • The cycling was added in methodology;
  • While editing the submission, the italics were accidentally removed. All the species have been checked and are now in italics.
  • The first sentence of Results was excluded.
  • Deposit number of CHIOC: we will provide the numbers as soon as the paper is accepted.
  • “Placing both measures in the text generates confusion. Wouldn't it be better to put them in a table?” We prefer maintaining both measures along the text because it facilitates the understanding.
  • We included a Remarks, where the sentence “Comparative measurements of O. pedicathedrae from different host species are given in Table 1” and data comparing the morphometry of the present work with previous reports were added.
  • The drawn specimen was the one with the best visible morphological characters to be represented.
  • We agree in fusing Figs 2 and 3, and Figs 4 and 5.
  • The references listed in Table 1 were merged in the last column.
  • The second sentence of discussion was excluded. All suggestions of this section were accepted.
  • “What did you add to the morphology to justify the redescription?” Morphometry of the species on a new host and confirmation of the morphology of the ventral sucker are being provided. In addition, as new genetic sequences are being included, it is important to make a detailed redescription of the material studied for future comparisons.

Thank you for your attention.

Simone C Cohen and co-authors

Reviewer 2 Report

Comments and Suggestions for Authors

This is a useful paper, adding further data to a poorly known, but huge, family. There is evidence that the writing has been hurried and the coverage of the literature is inadequate.

Throughout the paper all of  generic and specific names should be italicized.

The first paragraph of the Results section is directly from the instructions to authors and should be omitted.

Description: The overall length measurements in square brackets are given in millimetres, whereas those not in brackets are given in micrometres.

Omit ‘the’ and ‘a’ from note style description.

‘Ventral sucker [VS] located at anterior third of body, pedunculated, roughly half to third size of the oral sucker.’ In fact the ventral sucker is much larger than the oral sucker.

Figure 3. Opecoeloides pedicathedrae … (vf) reaching to the end of the body.

Opecoeloides pedicathedrae and O. fimbriatus are widely distributed in the Atlantic coast of and the Isthmus of Panama [3, 7, 40, 23, 22], while the hosts of O. furcatus were only reported from Mediterranean waters [41, 21].’ This statement needs to be corrected and more relevant references cited. Refer to the World Register of Marine Species.

O. fimbriatus is reported from Chamela, in the province of Jalisco on the Pacific coast of Mexico by Perez-Ponce de León et al. (1999) and Pérez-Ponce de León & Hernández-Mena (2019). The latter authors provide a 28S rDNA sequence for their worm (MK648309) which should be included in the analysis particularly as the sequence from Andres et al. (2014) is from the Atlantic basin.

O. furcatus is reported from off the coast of Belgium by van Beneden (1870), from Naha, Okinawa Prefecture, Japan by Yamaguti (1942) and off the Canary Islands by Gijon-Botella & Lopez-Roman (1989).

Gijon-Botella, H., Lopez-Roman, R. (1989). Aportacion al catalogo de Digenea de peces marinos del Archipelago de Canarias. Revista Ibérica de Parasitología. 49: 137–138.

Pérez-Ponce de León, G., García-Prieto, L., Mendoza-Garfias, B., León-Règagnon, V., Pulido-Flores, G., Aranda-Cruz, C., García-Vargas, F. (1999). Listados Faunísticos de México IX. Biodiversidad de Helmintos parásitos de peces marinos y estuarinos de la Bahía de Chamela, Jalisco. Instituto de Biología, Universidad Nacional Autónoma de México, Mexico City, 51 pp.

Pérez-Ponce de León, G.; Hernández-Mena, D. (2019). Testing the higher-level phylogenetic classification of Digenea (Platyhelminthes, Trematoda) based on nuclear rDNA sequences before entering the age of the 'next-generation' Tree of Life. Journal of Helminthology. 93 (3): 260–276.

van Beneden, P.J. (1870). Les poissons des côtes de Belgiques, leurs parasites et leurs commensaux. Mémoires de l'Académie Royal des Sciences, des Lettres et des Beaux-Arts de Belgique. 38: i–xx; 1–100.

Yamaguti, S. (1942). Studies on the helminth fauna of Japan. Part 39. Trematodes of fishes mainly from Naha. Transactions of the Biogeographical Society of Japan. 3: 329-398.

Comments on the Quality of English Language

A minor edit would improve the readability, but the paper is understandable as it is.

Author Response

Answer to Reviewer 2

We would like to thank the reviewers for their valuable suggestions. The following changes were made according to their suggestions on the manuscript “Redescription and first nucleotide sequences of Opecoeloides pedicathedrae (Digenea: Opecoelidae), a parasite of Cynoscion leiarchus (Cuvier, 1830) (Eupercaria: Sciaenidae) from Brazil”. Most suggestions were accepted, and the exceptions are explained. A “Remarks” was added as suggested by Reviewer 1.

We have added the sequence of O. fimbriatus (MK648309) to the 28S phylogenetic tree as suggested by Reviewer 2, then some information has been added to the results and this sequence has been included in table 2. The suggested references were included.

REV 2

  • “Throughout the paper all of generic and specific names should be italicized” - While editing the submission, the italics were accidentally removed. All the species have been checked and are now in italics.
  • The first paragraph of the Results section was deleted.
  • “Description: The overall length measurements in square brackets are given in millimeters, whereas those not in brackets are given in micrometers.” – Corrected.
  • Omit ‘the’ and ‘a’ from note style description – Done.
  • “‘Ventral sucker [VS] located at anterior third of body, pedunculated, roughly half to third size of the oral sucker.’ In fact, the ventral sucker is much larger than the oral sucker.” – Corrected.
  • The legend of Figure 3 was corrected.
  • The discussion with additional locality data from Opecoeloides fimbriatus and Opecoeloides furcatus has been improved. The suggested references have been included.
  • The 28S rDNA sequence of Opecoeloides fimbriatus (MK648309) reported from Pacific coast of Mexico by Pérez-Ponce de León & Hernández-Mena (2019) were included in the present study. The discussion with additional locality data from Opecoeloides fimbriatus and Opecoeloides furcatus has been improved. The suggested references have been included.

Thank you for your attention.

Simone C Cohen and co-authors

Round 2

Reviewer 2 Report

Comments and Suggestions for Authors

The form sequenced by Constenla et al. (2011) under the name Bathycreadium elongata were considered a different species by Perez-del-Olmo et al. (2014) and named B. brayi Pérez-del-Olmo, Dallarés, Carrassón & Kostadinova, 2014. This has been recognised by later authors, for example Martin et al. (2019).

Pérez-Del-Olmo, A.; Dallarés, S.; Carrassón, M.; Kostadinova, A. (2014). A new species of Bathycreadium Kabata, 1961 (Digenea: Opecoelidae) from Phycis blennoides (Brünnich) (Gadiformes: Phycidae) in the western Mediterranean. Systematic Parasitology. 88(3): 233-244., available online at https://doi.org/10.1007/s11230-014-9491-6